# Dual-Functionalized Pesticide Nanocapsule Delivery System with Improved Spreading Behavior and Enhanced Bioactivity

**DOI:** 10.3390/nano10020220

**Published:** 2020-01-27

**Authors:** Jianxia Cui, Changjiao Sun, Anqi Wang, Yan Wang, Huaxin Zhu, Yue Shen, Ningjun Li, Xiang Zhao, Bo Cui, Chong Wang, Fei Gao, Zhanghua Zeng, Haixin Cui

**Affiliations:** Institute of Environment and Sustainable Development in Agriculture, Chinese Academy of Agricultural Sciences, Beijing 100081, China; jianxiacui6075@163.com (J.C.); sunchangjiao@caas.cn (C.S.); angelking521@163.com (A.W.); zhuhuaxin0901@163.com (H.Z.); shenyue@caas.cn (Y.S.); liningjun09@126.com (N.L.); zhaoxiang@caas.cn (X.Z.); cuibo@caas.cn (B.C.); wangchong01@caas.cn (C.W.); gaofei@caas.cn (F.G.); zengzhanghua@caas.cn (Z.Z.)

**Keywords:** validamycin, thifluzamide, dual-functionalized pesticide nanocapsules, storage stability, foliar spread, bioactivity, synergistic effect

## Abstract

The prevention and control of pests and diseases are becoming increasingly difficult owing to extensive pesticide resistance. The synergistic use of pesticides for disease control is an effective way of slowing pesticide resistance, reducing the number of pesticide applications, and protecting the environment. In this study, a dual-functionalized pesticide nanocapsule delivery system loaded with two active ingredients (AIs)—validamycin and thifluzamide—was developed to prevent and control rice sheath blight; the nanocapsule system was based on a water–oil–water double emulsion method combined with high-pressure homogenization technology. Our results showed that the dual-functionalized pesticide nanocapsules were monodisperse spheres with a mean particle size of ~260 nm and had good storage stability. Compared with commercial formulations, the dual-functionalized pesticide nanocapsules exhibited good foliar spread owing to their small size, which is beneficial for reducing the loss of pesticides on the leaves. The 50% median effect concentration and synergistic ratio against *Rhizoctonia solani* of the dual-functionalized pesticide nanocapsules and commercial formulation were 0.0082 and 0.0350 μg/mL, and 2.088 and 0.917, respectively. These findings indicate that the bioactivity of the dual-functionalized system was significantly better than that of the commercial formulations and that the dual-functionalized system demonstrated a clear synergistic effect between the two AIs. The system presented here is simple, fast, and capable of dual-pesticide loading with significant synergistic effects. Our findings could help to facilitate the improvement of pesticides efficiency and the slowing of pesticide resistance.

## 1. Introduction

Sheath blight is one of the most harmful diseases in rice (*Oryza sativa* L.) worldwide, and is the most prevalent among the three major rice diseases in China [1,2]. For several decades, the control of rice sheath blight has relied primarily on the use of validamycin. However, due to the emergence of pesticide resistance, the control efficiency of validamycin has declined in recent years [3,4]. In addition, when conventional pesticide formulations are used to control rice sheath blight, their active ingredients (AIs) can be reduced as a result of spray drift, run-off, and degradation during field application [5,6,7,8,9,10]. Owing to their low efficiency, the extensive use of conventional pesticide formulations has had deleterious effects such as environmental pollution, threats to non-target organisms, and pesticide resistance; these cumulative effects significantly aggravate food safety issues, posing a significant threats to human health [11,12,13,14,15].

The modification of existing pesticides and the development of novel pesticides with high control efficiency and low toxicity is essential to slow the progression of pesticide resistance [16]. However, the development of new AI is becoming increasingly difficult, making it more desirable to formulate complex pesticide formulations through precise selection and targeted combination of AIs based on their synergistic effects [17]. This provides an effective approach for improving the utilization efficiency of existing pesticides, delaying the occurrence of pathogen resistance, broadening the spectrum of disease control, and prolonging the service life of pesticide formulations. However, the complex formulations available on the market are simply mixtures of existing pesticide formulations. Although these formulations can slow the occurrence of pesticide resistance to a certain extent, unsolved problems remain, such as the requirement for large amounts of organic solvents, drift, and poor dispersity in water [18,19]. Furthermore, conventional pesticide formulations always rapidly fall below the effective concentration level due to hydrolysis, photolysis or microbial degradation [19]. A delivery system could effectively prevent the premature degradation of pesticides, achieve continuous and stable release of AIs, and maintain a predetermined minimum effective level of pesticides for a suitable time period [20].

Encapsulation technology has attracted emerging interest. By encapsulating liquid or solid pesticides into the shell material, capsules could protect AIs from the degradation caused by environmental factors, which provides a new strategy for improving the utilization rate of pesticides [21,22,23,24]. Nanocapsules can also reduce the loss of pesticides to non-target environments, achieve sustainable release of AIs, and maintain effective control concentrations over extended periods of time [8,25,26,27,28,29]. However, most of the available pesticide capsules are on the micron scale [30]. By comparison, nanocapsules demonstrate clear advantages for improving the foliar deposition and spread of pesticides and enhancing their bioactivity due to their small-size and large specific surface area [31,32,33,34,35]. The preparation of nanocapsules requires strict conditions [36,37,38,39], however, research on dual-pesticide nanocapsules is relatively scarce.

Thifluzamide is a novel and highly effective systemic fungicide that is used for the control of rice sheath blight [40,41,42]. In this study, thifluzamide and validamycin were selected as a synergistic combination. Dual-functionalized pesticide nanocapsules were prepared based on the distinct physical and chemical properties of the selected pesticides. Our findings demonstrate that the ratio of the pesticides, the physicochemical properties, and the morphology, particle size, and structure of the dual-functionalized pesticide nanocapsule delivery system can be adjusted to improve the field control efficiency of the pesticides. This control of pesticide application slowed the occurrence of pathogen resistance, broadened the microbicidal spectrum, and reduced the cost of pesticide application.

## 2. Materials and Methods

### 2.1. Materials

Validamycin A (60%) and thifluzamide (95%) were purchased from Bailexin Biotech. Co., Ltd. (Beijing, China). A validamycin standard was obtained from Leboward Tech. Co., Ltd. (Beijing, China) and a thifluzamide standard was supplied by Bailinwei Tech. Co., Ltd. (Beijing, China). Validamycin wettable powder (VWP) and aqueous solution (VAS) were supplied by Huifeng Biosciences Co., Ltd. (Tonglu, China) and Lvchuan Biotech. Industry Co. Ltd. (Fuzhou, China), respectively. The thifluzamide suspension concentrate (TSC) was manufactured by Limin Agrochemical Co., Ltd. (Yancheng, China), and the validamycin/thifluzamide suspension concentrate (VTSC) was supplied by Nannong Pesticide Technology Development Co., Ltd. (Nanjing, China).

Polylactide (PLA, Mw ~ 100 KDa) was supplied by Nature Works Co. (Minnetonka, MN, USA). Polyvinyl alcohol (PVA), 87%–90% hydrolyzed with an average Mw of 30,000–70,000, was supplied by Sigma-Aldrich Shanghai Trading Co., Ltd. (Shanghai, China). Styryl phenol polyoxyethylene ether (emulsifier 600#), alkylphenol formaldehyde resin polyoxyethylene ether (emulsifier 700#), and polyoxyethylene castor oil ether (EL-40) were purchased from Cangzhou Hongyuan Agrochemical Co., Ltd. (Cangzhou, China). Polyoxyethylene sorbitan monooleate (Tween 80) was purchased from J&K Chemical (Beijing, China). Maleic rosinpolyoxypropylene-polyoxyethylene ether sulfonate (MRES) and polycarboxylate (PC) were provided by Sinvochem S&D Co., Ltd. (Jiangsu, China). Octylphenol polyoxyethylene ether-10 (OP-10) was purchased from Jinyue Biotech. Co., Ltd. (Beijing, China). Sodium dodecyl sulfate (SDS) was supplied by BioMS Biotech. Co., Ltd. (Beijing, China). Castor oil polyoxyethylene ether-125 (BY-125) was obtained from Jiangsu Zhongshan Chemical Co., Ltd. (Nanjing, China).

Analytical-grade dichloromethane and chloroform were obtained from Beijing Chemical Works (Beijing, China). Acetonitrile and methanol (high-performance liquid chromatography (HPLC) grade) were purchased from Fisher (Shanghai, China). The water used in all experiments was purified using a Milli-Q water purification system (Dubuque, IA, USA).

### 2.2. Methods

#### 2.2.1. Preparation of Dual-Functionalized Pesticide Nanocapsules

The dual-functionalized pesticide nanocapsule delivery system was prepared using a water–oil–water (W/O/W) double emulsion method combined with high pressure homogenization technology. Briefly, 2.5 g of validamycin was dissolved in 15 mL of water to form an aqueous solution as the internal water phase. Next, 0.5 g of thifluzamide was dissolved in 60 mL of dichloromethane, followed by the addition of 3 g of PLA (capsule wall material) to form the oil phase. The external water phase was obtained by dissolving 4 g of surfactant in 300 mL of water. The internal water phase was dispersed in the oil phase using an ultrasonic homogenizer (JY92-IIN, Sientz, Ningbo, China) at 585 W to form a primary W/O emulsion, which was then added to the external water phase and sheared for 5 min at 19,000 rpm to prepare a coarse W/O/W double emulsion. To reduce the size of the emulsion particles, the coarse emulsion was transferred to a high-pressure homogenizer (ATS, AH-100D, Engineer Inc., Brampton, Canada) and treated three times under successive pressures of 300, 600, and 900 kPa to yield a fine double emulsion.

The obtained emulsion was mechanically stirred at 800 rpm for 20 h at room temperature using an electric stirrer (IKA, EUROSTAR 60, Staufen, Germany) to allow for solidification into capsules and eliminate excess organic solvents. The nanocapsules were collected by centrifugation at 10,000 rpm for 10 min at room temperature (Thermo SCIENTIFIC, ST 16R, Pittsburgh, PA, USA), and the precipitate was resuspended in deionized water. This process was repeated three times to remove excess impurities. Then, the precipitate was collected and frozen in an ultra-low-temperature freezer (Haier, DW-86W100, Qingdao, China). Excess water was removed using a freeze dryer (FD-81, ETELA, Tokyo, Japan).

#### 2.2.2. Particle Size and Morphological Characterization of Dual-Functionalized Pesticide Nanocapsules

A small portion of the solid powder was diluted in deionized water to obtain a transparent suspension. The mean particle size (MPS) and polydispersity index (PDI) of the nanocapsules in the transparent solution were measured by dynamic light scattering (DLS Zetasizer Nano ZS90, Malvern Instruments, Worcestershire, UK). Each sample was measured in parallel three times, and the mean values and standard deviation were calculated.

A scanning electron microscope (SEM, HITACHI, Su-8010, Tokyo, Japan) was used to characterize the morphology of the nanocapsules. An aliquot of the resuspended nanocapsules was dropped onto the surface of a cleaned silicon water. The SEM images were acquired at an acceleration voltage of 5 kV. A transmission electron microscope (TEM, HITACHI, HT7700, Tokyo, Japan) was used to characterize the internal structure of the prepared nanocapsules. An appropriate volume of the transparent solution was dropped onto 300-mesh copper grid coated with a carbon film. TEM images were acquired at an acceleration voltage of 80 kV.

#### 2.2.3. Determination of Dual-Functionalized Pesticide Nanocapsules’ Loading Efficiencies

The loading efficiency of the two AIs into the nanocapsules was determined by destruction of the nanocapsules. An appropriate aliquot sample was accurately weighed by an ultramicro analytical balance (METTLER TOLEDO, Zurich, Switzerland) and completely dissolved in chloroform to break the capsule wall. Excess solvent was removed by distillation under reduced pressure using a rotary evaporator (RE100-Pro, SCILOGEX, Rocky Hill, CT, USA). We then used 90% methanol to extract the pesticides before the dissolved sample of the two AIs was passed through a 0.22 μm membrane filter. The pesticide concentrations in the filtrate were determined by HPLC (1260 Infinity, Agilent Technologies, Palo Alto, CA, USA) using a C_18_ column (5 µm, 4.6 mm × 150 mm, Agilent Technologies; Santa Clara, CA, USA) at room temperature. The mobile phase for validamycin was 0.005 mol/L disodium hydrogen phosphate buffer solution (pH = 7.0), with a flow rate of 0.8 mL/min and a detection wavelength of 210 nm; the mobile phase for thiafuramide was acetonitrile/water (70:30) with a flow rate of 1 mL/min and a detection wavelength of 225 nm.

The loading efficiency of the nanocapsules was calculated as follows:

Pesticide loading rate (%) = effective mass of pesticide in nanocapsules/total mass of nanocapsules × 100.

#### 2.2.4. Stability Evaluation of Dual-Functionalized Pesticide Nanocapsules

The stability of the dual-functionalized pesticide nanocapsules was evaluated according to the General Principles of Pesticide Storage Stability Test at Normal Temperature (NY/T 1427–2016), Pesticide Thermal Storage Stability Test Method (GB/T 19136–2003), and Pesticide Cold Storage Stability Test Method (GB/T 19137–2003). Specifically, three parallel samples of the transparent solution were transferred into ground–glass tubes and sealed and stored separately at 0 ± 2 °C for 7 days and 25 ± 2 °C and 54 ± 2 °C for 14 days. Samples were removed at specific time intervals for stability evaluation based on MPS and PDI measurements.

#### 2.2.5. Foliar Spread and Contact Angle Evaluation of Dual-Functionalized Pesticide Nanocapsules

Cucumber (*Cucumis sativus* L.)—the chosen plant model—was grown in an artificial climate incubator (430D, Ningbo Jiangnan Instrument, Ningbo, China). Nanocapsules with a concentration of 200 μg/mL were sprayed onto the surface of the cucumber leaves and allowed to air-dry at room temperature. Control groups were treated with commercially available VTSC, TSC, VAS, and VWP. The dispersion of different samples on the foliage was observed using an environmental scanning electron microscope (ESEM; Quanta FEG 250, Houston, TX, USA).

To evaluate the contact angle, 7 μL of 200 μg/mL nanocapsule solution was dropped onto the surface of fresh leaves at room temperature using a microliter syringe. Images of each droplet were immediately captured using a contact angle goniometer (JC2000D2M; Zhongchen Digital Technology Equipment Co., Ltd., Shanghai, China). Each trial was repeated five times.

#### 2.2.6. Bioactivity Evaluation of Dual-Functionalized Pesticide Nanocapsules

*Rhizoctonia solani*, the pathogen responsible for rice sheath blight, was selected as a model fungus to evaluate the bioactivity of the nanocapsules. Commercial VTSC, VWP, and TSC were used as controls and sterile deionized water was used as the blank control. Briefly, a 5-mm-diameter punch was used to cut mycelium plugs from an activated culture of *R. solani* that grew uniformly. The plugs were inoculated onto the center of 90 mm-diameter potato dextrose agar plates. The plates were incubated at 25 °C for 36 h, and three parallel independent trials were performed for each concentration. Colony diameter was measured using the cross method and statistically analyzed. Growth inhibition rate and synergistic ratio (SR) were calculated. In addition, the toxicity regression equation and median effect concentration (EC_50_) were analyzed using Data Processing System (DPS, v7.05) [43].

The growth inhibition rate was calculated as follows:I% = (D_c_ − D_t_)/(D_c_ − D_d_) × 100%(1)
where I% is the growth inhibition rate of *R. solani*; D_c_ and D_t_ are the growth diameter of *R. solani* in the blank control and the treatment group, respectively; and D_d_ is the diameter of the mycelium plug (5 mm).

The interaction of the mixed pesticides was evaluated according to the Wadley method using the following equations:EC_50_ (theoretical value) = (a + b)/(a/EC_50a_ + b/EC_50b_)(2)
SR = EC_50_ (theoretical value)/EC_50_ (actual value)(3)
where a and b are the proportions of the two pesticides in the mixture. The effect of the mixture was analyzed based on the SR value: SR ≤ 0.5 indicates an antagonistic effect; SR = 0.5–1.5 indicates an additive effect; and SR ≥ 1.5 indicates a synergistic effect between the two pesticides [44].

#### 2.2.7. Statistical Analysis

Statistical analysis of the experimental data was carried out using DPS. The results are presented as “average value ± standard deviation”. The multiple comparative analysis was performed using the least significant difference (LSD) method. (*) *P* < 0.05, was regarded a significant difference between experimental groups.

## 3. Results and Discussion

### 3.1. Preparation of Dual-Functionalized Pesticide Nanocapsules

PLA, a safe, degradable, and low-cost polymer was selected as the capsule material for this study. Multiple emulsification methods including ultrasonication, shearing, and high-pressure homogenization, were adopted to prepare a delivery system with small particle size that was simultaneously loaded with two pesticides AIs (Figure 1). PVA and polycarboxylate were selected as surfactants to stabilize the nanocapsules. PVA, a non-ionic surfactant, formed an adsorptive layer to produce a steric hindrance effect to prevent nanocapsules’ aggregation [45]. Polycarboxylate, an anionic surfactant, provided sufficient electrical charge to increase the electrostatic repulsion between the nanocapsules [46]. The simultaneous use of surfactants with different properties improved the stability of the dual-functionalized pesticide nanocapsules.

### 3.2. Effects of Surfactants on the Particle Size and Polydispersity of Dual-Functionalized Pesticide Nanocapsules

#### 3.2.1. Single Surfactants

MPS and PDI are two important indicators used to evaluate the performance of dual-functionalized pesticide nanocapsules. Figure 2 summarizes the MPS and PDI of the nanocapsules prepared with different single surfactants. A total of 10 typical surfactants were selected to explore their effects on nanocapsule synthesis, and all 10 surfactants had significant effects on the MPS and PDI of the nanocapsules. The MPS and PDI of the nanocapsules prepared with MRES, PVA, SDS, and PC were all less than 300 nm and 0.3, respectively, indicating good dispersity and small nanocapsule size in the presence of these four surfactants.

It was found that the emulsification effect was poor for the nanoparticles prepared with SDS, which produced a coarse unstable emulsion. Therefore, the morphology of the particles prepared with MRES, PVA, and PC was characterized by SEM. Although MRES facilitated the formation of small particles with good dispersity, their morphology did not show uniformly stable spheres (Figure 3), however both PVA and PC supported the formation of regular spheres. Based on these results, the combination of PVA and PC was preferentially selected for the preparation of the dual-pesticide nanocapsules.

#### 3.2.2. Complex Surfactants

Surfactants are substances composed of hydrophilic and hydrophobic groups that can substantially reduce the surface tension of a solution [47]. In recent years, surfactants have commonly been mixed to further improve their performance by taking advantage of their synergistic effects [48]. Correct surfactant content plays an important role in stabilizing nanoparticles [49,50]. In the current study we mixed the two selected surfactants that demonstrated good emulsification effects and tested their performance in different ratios (PVA: PC = 1:1, 1:2, 2:1, 1:3, and 3:1). The MPS, diameters based on 90% of the cumulative distribution profile (D90), and PDI of the nanocapsules obtained for the different ratios are presented in Table 1. All PVA:PC ratios yielded nanocapsules with uniform size within the range 250–300 nm. For each ratio, the PDI was below 0.3, which implied good nanocapsules’ dispersity [51]. The use of surfactant combinations showed the potential to yield nanocapsules with smaller particle size and better physical stability [52].

### 3.3. Morphology of Dual-Functionalized Pesticide Nanocapsules

The nanocapsules prepared with different surfactant ratios were characterized by TEM, which showed small spherical nanocapsules with regular morphology and distribution (Figure 4). Varying the surfactant ratio did not significantly affect the nanocapsules in terms of their particle size, dispersity, or morphology. However, under the same conditions, the solid precipitate collected with a PVA:PC ratio of 2:1 had a more uniform and smaller particle size. Thus, a PVA: PC surfactant ratio of 2:1 was selected to prepare samples for subsequent characterization.

### 3.4. Stability of Dual-Functionalized Pesticide Nanocapsules

The storage stability of pesticides is an important indicator in the evaluation of pesticide systems [53]. The MPS and PDI of the nanocapsules after storage in various conditions were measured to evaluate the physical stability of the complex pesticide. After 7 days of storage at 0 °C, the MPS decreased from 291.7 to 289.7 nm. After 14 days of storage at 25 °C and 54 °C, the MPS decreased from 291.7 nm to 284.5 and 263.7 nm, respectively (Figure 5). The MPS remained relatively stable without significant changes throughout the storage process. In addition the PDIs of the nanocapsule suspensions were below 0.3, indicating a uniform size distribution and good dispersity of the nanocapsules [51].

When stored at 0 °C and 25 °C, the MPS and PDI of the nanocapsule suspensions remained relatively stable, implying that the dual-functionalized pesticide nanocapsules had the desired physical stability. However, storage at 54 °C accelerated the process of the dual-functionalized pesticide nanocapsules approaching the glass transition temperature of PLA, which caused PLA to change from a rigid state to a flexible state [54,55], resulting in a reduction in particle size. However, the pesticide samples are not expected to experience such high temperatures under normal circumstances, so the prepared nanocapsules are still regarded as having good storage stability.

### 3.5. Foliar Spread and Contact Angle Evaluation of Dual-Functionalized Pesticide Nanocapsules

The foliar spread and contact angle of the dual–functionalized pesticide nanocapsules on plant leaves were characterized using an ESEM and a contact angle goniometer (Figure 6). The contact angles of the blank control, dual-functionalized pesticide nanocapsules, and commercial VTSC, TSC, VAS, and VWP on cucumber leaves were 94.97°, 77.10°, 69.7°, 92.36°, 89.62°, and 77.96°, respectively. According to these data, the prepared nanocapsules showed no clear advantage compared with other groups. This observation is likely due to the fact that no additives such as sizing and spreading agent were added to the preparation of dual-functionalized pesticide nanocapsules [56]. However, the ESEM images clearly show that the commercial formulations were spread on the leaf surface in patches, whereas the dual-functionalized pesticide nanocapsules were scattered on the leaf surface as uniform spheres. Compared with commercial formulations, the dual-functionalized pesticide nanocapsules improved the foliar spreading behavior. As a result of the unique micro-nano structure, the nanocapsules were able to embed into the leaf vein structure [9], which prevented the nanocapsules from collapsing and promoted pesticide absorption and transportation by the leaf. These qualities are, thereby, expected to effectively increase the utilization efficiency of the pesticides and consequently lead to a reduction of pesticide application dosage.

### 3.6. Bioactivity of the Dual-Functionalized Pesticide Nanocapsules

The bioactivity of dual–functionalized pesticide nanocapsules was evaluated to verify the feasiblility of applying the new pesticide formulation. The loading efficiencies for validamycin and thifluzamide in the nanocapsules were 9.9% and 15.8%, respectively. The colony growth of *R. solani* treated with the same concentration of AIs is shown in Figure 7. As shown in Table 2, the EC_50_ of VWP, TSC, the dual-functionalized pesticide nanocapsules, and VTSC were 227.5372, 0.0107, 0.0082, and 0.0350 μg/mL, respectively. These results demonstrate that the dual-functionalized pesticide nanocapsules had a synergistic effect compared with VWP and TSC, and the toxicity of the nanocapsules was 4.2 times greater than that of VTSC. In addition, the SR values for the dual-functionalized nanocapsuled and VTSC were 2.088 and 0.917, respectively. These results indicate that the two AIs in the dual-functionalized pesticide nanocapsules had a significant synergistic effect, whereas VTSC alone exhibited an additive effect between the two AIs. The nanocapsules were, therefore, superior to the commercial control in terms of both relative toxicity and synergistic effect; which is attributed to the unique micro-nano structure being able to facilitate adsorption and transportation into fungal cells [57,58]. Therefore, the nanocapsules substantially improved the antifungal activity of the pesticides, which could potentially reduce the pesticide application rate.

## 4. Conclusions

In this study, nanocapsules loaded with two pesticides—thifluzamide and validamycin—were successfully prepared based on a double emulsion method combined with high-pressure homogenization technology. The effects of single surfactants and the mixing ratio of complex surfactants on the dual-functionalized pesticide nanocapsules were investigated. The particle size, morphology, loading efficiency, stability, foliar spread, and bioactivity of the nanocapsules were also analyzed. The, nanocapsules showed better spreading performance on foliage compared with commercial pesticide formulations. Compared with the commercial formulation, the dual-functionalized pesticide nanocapsule delivery system demonstrated a synergistic effect, which significantly enhanced the bioactivity against *R. solani*. This technology is expected to slow down the widespread pesticide resistance problems by improving the effective utilization rate of pesticides while reducing the pesticide input. In addition, this study provides new strategies for the further development of new pesticide formulations. Future attempts will be addressed to the large–scale production of the dual-functionalized pesticide nanocapsule delivery system, which could facilitate the industrial application of this technology.

## Figures and Tables

**Figure 1 nanomaterials-10-00220-f001:**
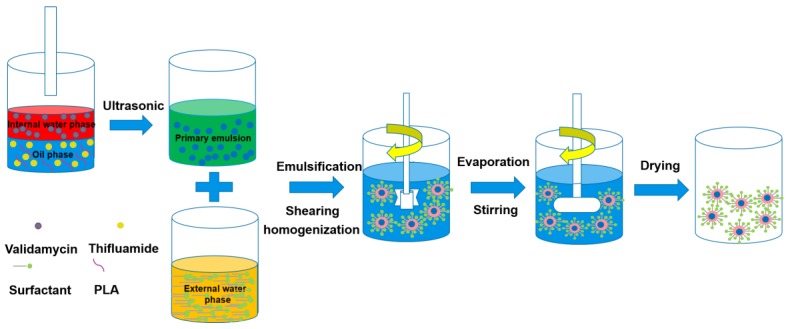
Schematic illustration of the preparation of dual-functionalized pesticide nanocapsules.

**Figure 2 nanomaterials-10-00220-f002:**
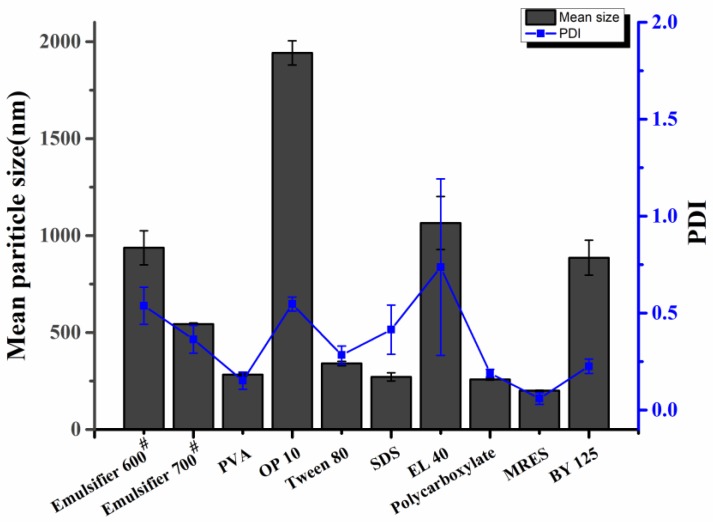
Effects of single surfactants on the mean particle size (MPS) and polydispersity index (PDI) of the nanocapsules.

**Figure 3 nanomaterials-10-00220-f003:**

Scanning electron microscopy (SEM) morphologies of the nanocapsules (**A**) maleic rosinpolyoxypropylene-polyoxyethylene ether sulfonate (MRES); (**B**) polyvinyl alcohol (PVA); and (**C**) polycarboxylate (PC)).

**Figure 4 nanomaterials-10-00220-f004:**
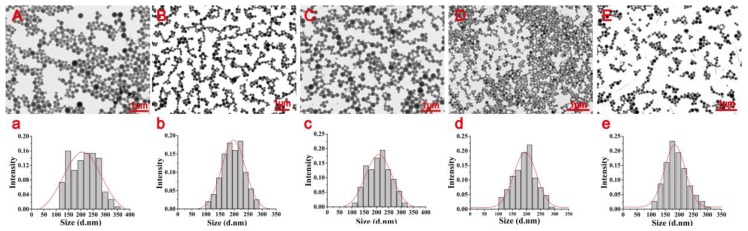
Transmission electron microscopy (TEM) morphologies of the nanocapsules (**A**–**E**) and statistical dried particle size distributions of nanocapsules (**a**–**e**) based on the TEM images. (**A**–**a**, **B**–**b**, **C**–**c**, **D**–**d**, and **E**–**e** represent PVA: PC = 1:1, 1:2, 2:1, 1:3, and 3:1, respectively).

**Figure 5 nanomaterials-10-00220-f005:**
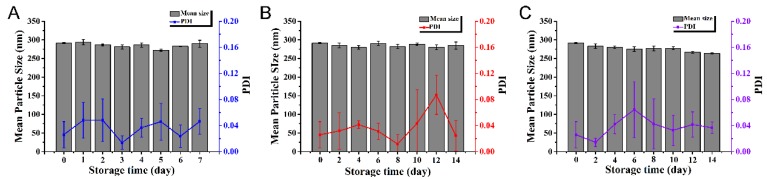
Effects of temperature on the MPS and PDI of the nanocapslues. (**A**) 0 °C; (**B**) 25 °C; and (**C**) 54 °C.

**Figure 6 nanomaterials-10-00220-f006:**
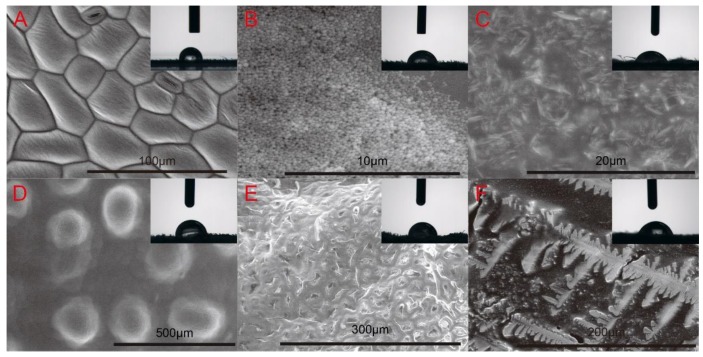
Foliar spread and contact angle of different pesticide formulations on cucumber leaves (**A**) blank control; (**B**) dual-functionalized pesticide nanocapsules; (**C**) VTSC; (**D**) TSC; (**E**) VAS; (**F**) VWP.

**Figure 7 nanomaterials-10-00220-f007:**
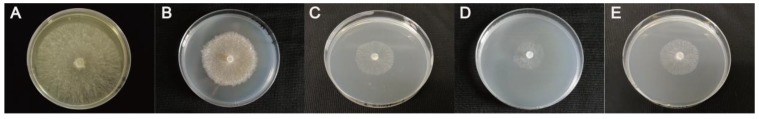
Colony growth of *Rhizoctonia solani* under treatment with different pesticide preparations (**A**) blank control; (**B**) VWP; (**C**) TSC; (**D**) dual-functionalized pesticide nanocapsules; and (**E**) VTSC.

**Table 1 nanomaterials-10-00220-t001:** Effects of complex surfactants on the MPS, D90, and PDI of the nanocapsules.

PVA:PC	MPS (d.nm)	D90 (nm)	PDI
1:1	287.4 ± 0.7	373.3 ± 26.3	0.051 ± 0.046
1:2	265.3 ± 0.8	330.7 ± 6.0	0.034 ± 0.030
2:1	279.7 ± 0.2	359.7 ± 14.5	0.042 ± 0.002
1:3	282.4 ± 2.7	378.7 ± 8.0	0.043 ± 0.018
3:1	288.2 ± 3.7	355.7 ± 8.7	0.025 ± 0.017

**Table 2 nanomaterials-10-00220-t002:** Bioactivity of the dual-functionalized pesticide nanocapsules against *Rhizoctonia solani.*

Treatments	Toxicity Regression Equation	Correlation Coefficient (R^2^)	36-h EC_50_ (μg/mL) (95% CI)	EC_50_ (μg/mL) (Theoretical)	Synergistic Ratio
VWP	Y = 0.8485x + 3.0276	0.9716	227.5372 (135.8436–339.1162)	–	–
TSC	Y = 0.5736x + 6.1966	0.9802	0.0107 (0.0050–0.0222)	–	–
Dual–functionalized pesticide nanocapsules	Y = 0.5736x + 6.1966	0.9974	0.0082 (0.0062–0.0110)	0.01712	2.088
VTSC	Y = 0.5654x + 5.8325	0.9683	0.0350 (0.0176–0.0992)	0.03210	0.917

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
