# Peer review of "Dual-Functionalized Pesticide Nanocapsule Delivery System with Improved Spreading Behavior and Enhanced Bioactivity"

_nanomaterials, 2020, doi:10.3390/nano10020220_

Round 1

Reviewer 1 Report

The authors describe a new delivery system of pesticide; the nanocapsules led to an improvement the spreading behavior and the bioactivity.

The manuscript is well written in good English and in an adequate format for Nanomaterials. Typing or syntax errors have to be corrected by a careful revision.

The introduction is devoted to existing pesticides and to the development of novel ones. The references are for most of them recent and appropriate. However, the part devoted to the “classical” pesticides has to be developed in order to explain the attempted effect of the encapsulation. This part has to be revised.

Next, the experimental part describing the formation of the nanocapsules and their analysis is quite clear. These ones have been successfully performed through a double emulsion method combined with high pressure homogenization technology. This part is well described and illustrated.

Furthermore, the effects of single surfactants and the mixing ratio of complex surfactants on the dual-functionalized pesticide nanocapsules have been carefully studied and analyzed.

The analytic information concerning the size of the particle, their morphology, their loading efficiency and stability led to the choice of specific compounds. The foliar spread and the bioactivity of the nanocapsules have also been studied and revealed the interest of the encapsulation.

Moreover, It could be interesting to improve this kind of technology on a wider scale. Have the authors tried to extend this technology?

In conclusion, considering these remarks and comments, I could recommend this manuscript for a publication in Nanomaterials but after major revisions.

Author Response

Response to reviewer’s comments

Dear reviewer,

We very much appreciate the careful reading of our manuscript and the valuable suggestions. The manuscript is revised submission with new line in the text. The major revised portions are marked in red. We respond point by point to each comment as listed below, along with a clear indication of the location of the revision.  Hope these will make it more acceptable for publication. 

Point 1: The manuscript is well written in good English and in an adequate format for Nanomaterials. Typing or syntax errors have to be corrected by a careful revision.

Response 1: Thanks a lot for the reviewer’s useful comments. Some typing and syntax errors have been carefully corrected by the native English speaker. And these changes are clearly visible in the updated manuscript.

Point 2: The introduction is devoted to existing pesticides and to the development of novel ones. The references are for most of them recent and appropriate. However, the part devoted to the “classical” pesticides has to be developed in order to explain the attempted effect of the encapsulation. This part has to be revised.

Response 2: According to the rigorous opinions of the reviewers, the introduction part has been improved and seven extra references have been cited to explain the attempted effect of the encapsulation.

Point 3: Next, the experimental part describing the formation of the nanocapsules and their analysis is quite clear. These ones have been successfully performed through a double emulsion method combined with high pressure homogenization technology. This part is well described and illustrated.

Response 3: We appreciate Reviewer’s positive comments that “the experimental part describing the formation of the nanocapsules and their analysis is quite clear. This part is well described and illustrated.”

Point 4: Furthermore, the effects of single surfactants and the mixing ratio of complex surfactants on the dual-functionalized pesticide nanocapsules have been carefully studied and analyzed.

Response 4: We appreciate the reviewers for affirming our research on the effects of surfactants on dual-functionalized pesticide nanocapsules.

Point 5: The analytic information concerning the size of the particle, their morphology, their loading efficiency and stability led to the choice of specific compounds. The foliar spread and the bioactivity of the nanocapsules have also been studied and revealed the interest of the encapsulation.

Response 5: We appreciate the reviewers for their careful review of our research work and the concise summary of the research content.

Point 6: Moreover, it could be interesting to improve this kind of technology on a wider scale. Have the authors tried to extend this technology?

Response 6: Of course yes. Our study is mainly focused on connecting the selection of synergistic pesticides combinations with the preparation of nanocapsule delivery systems. This technology is expected to slow down the widespread pesticide resistance problems by improving the effective utilization rate of pesticides while reducing the pesticide input. This technology can be adopted when one of the active ingredients of the selected synergistic combination is hydrophilic and the other is hydrophobic. This study provides new strategies for the further development of new formulations. It is relatively easy to operate and corresponding patent has been applied. Besides, in our future research work, a lot of attempts will be addressed to the large-scaled production of the dual-functionalized pesticide nanocapsule delivery system, which could facilitate the industrial application of this technology.

Point 7: In conclusion, considering these remarks and comments, I could recommend this manuscript for a publication in Nanomaterials but after major revisions.

Response 7: We appreciate the reviewer for the recognition of our research work, and we will carefully revise the manuscript according to the reviewers' comments for publication.

Reviewer 2 Report

The work by Cui et al. (Dual-functionalized pesticide nanocapsule delivery system with improved spreading behavior and enhanced bioactivity) reports dual-functionalized pesticide nanocapsule delivery system loaded with two active ingredients (AIs) such as validamycin and thifluzamide. The proposed nanocapsule system was based on a water-oil-water double emulsion method combined with high pressure homogenization technology. With this approach, they fabricated monodispersed spheres with a mean particle size of ~260 nm and good storage stability. In comparison to commercial formulations, the bioactivity of the dual-functionalized system was significantly improved due to the synergistic effect between the two AIs.

The report is well-organised and well-written. The data presented in the manuscript provides valuable and novel information to the readers. I recommend the publication of the report after the reconsideration of the some minor points given below.

Line 40

First, define the full name of the AIs and then abbrevations.

Line 84

Please define the molecular weight of PLA.

Please refer to Figure 6 in the manuscript.

Line 323

Please reconsider this statement.

Please mention the Table 2 in the manuscript.

Author Response

Response to reviewer’s comments

Dear reviewer,

We very much appreciate the careful reading of our manuscript and the valuable suggestions. The manuscript is revised submission with new line in the text and the major revised portions are marked in red. We also respond point by point to each comment as listed below, along with a clear indication of the location of the revision.  Hope these will make it more acceptable for publication. 

Point 1: The work by Cui et al. (Dual-functionalized pesticide nanocapsule delivery system with improved spreading behavior and enhanced bioactivity) reports dual-functionalized pesticide nanocapsule delivery system loaded with two active ingredients (AIs) such as validamycin and thifluzamide. The proposed nanocapsule system was based on a water-oil-water double emulsion method combined with high pressure homogenization technology. With this approach, they fabricated monodispersed spheres with a mean particle size of ~260 nm and good storage stability. In comparison to commercial formulations, the bioactivity of the dual-functionalized system was significantly improved due to the synergistic effect between the two AIs.

Response 1: We appreciate the reviewer for the careful review of our manuscript as well as the generalization of our research methods and contents.

Point 2: The report is well-organized and well-written. The data presented in the manuscript provides valuable and novel information to the readers. I recommend the publication of the report after the reconsideration of the some minor points given below.

Response 2: We appreciate reviewer ’s positive comments that our report is “well-organized and well-written. The data presented in the manuscript provides valuable and novel information to the readers.” We will carefully correct the manuscript according to the reviewer's comments so that the manuscript can be ready for publication.

Point 3: Line 40: First, define the full name of the AIs and then abbreviations.

Response 3: We agree with the reviewer’s assessment and we have made corresponding changes in the manuscript. The full name of the AIs has been defined firstly and then abbreviated.

Point 4: Line 84: Please define the molecular weight of PLA.

Response 4: Based on the helpful comments of the reviewer, we have added the molecular weight of PLA in the “Materials and Methods” part.

Point 5: Please refer to Figure 6 in the manuscript.

Response 5: According to the rigorous opinions of the reviewers, we have marked the “Figure 6”in the appropriate place.

Point 6: Line 323: Please reconsider this statement;

Response 6: Thanks a lot for the reviewer’s useful comments. The statement has been changed as follows:

The dual-functionalized pesticide nanocapsule delivery system demonstrated a synergistic effect which significantly enhanced bioactivity than commercial pesticide. This technology is expected to slow down the widespread pesticide resistance problems by improving the effective utilization rate of pesticides while reducing the pesticide input. In addition, this study provides new strategies for the further development of new pesticide formulations. Future attempts will be addressed to the large-scaled production of the dual-functionalized pesticide nanocapsule delivery system, which could facilitate the industrial application of this technology.

Point 7: Please mention the Table 2 in the manuscript.

Response 7: Based on the helpful comments of the reviewer, table 2 was mentioned appropriately in the manuscript.

Round 2

Reviewer 1 Report

The reviewer thanks the author for improving seriously their manuscript.

No problem now for a publication of this revised form in Nanomaterials.